Fine-root traits of allelopathic rice at the seedling stage and their relationship with allelopathic potential

Li Jiayu 1 2
Lin Shunxian 1
Zhang Qingxu 1
Zhang Qi 1
Hu Wenwen 1
He Haibin ljy@fafu.edu.cn 1 2
1 Fujian Provincial Key Laboratory of Agroecological Processing and Safety Monitoring, College of Life Sciences, Fujian Agriculture and Forestry University , Fuzhou , Fujian , China
2 Key Laboratory of Crop Ecology and Molecular Physiology, Fujian Agriculture and Forestry University , Fuzhou , Fujian , China
Mysore Kirankumar
Electronic publication date: 2019 Jun 12
Publication date: 2019
Volume: 7
Electronic Location ID: e7006
Received 2018 Sep 21; Accepted 2019 Apr 22
Copyright: ©2019 Li et al.
Copyright year: 2019
Copyright holder: Li et al.
License: This is an open access article distributed under the terms of the Creative Commons Attribution License, which permits unrestricted use, distribution, reproduction and adaptation in any medium and for any purpose provided that it is properly attributed. For attribution, the original author(s), title, publication source (PeerJ) and either DOI or URL of the article must be cited.
License URL: https://creativecommons.org/licenses/by/4.0/

Keywords: Rice (oryza sativa), Allelopathy, Fine root traits, Root exudates, Phenolic acids

Funding: National Natural Science Foundation 31701802 31370380 Fujian-Taiwan Joint Innovative Center FJ 2011 Program, NO. 2015–75 This work was supported by grants from the National Natural Science Foundation (31701802, 31370380), and Fujian-Taiwan Joint Innovative Center for Germplasm Resources and Cultivation of Crop (FJ 2011 Program, NO.2015-75), China. The funders had no role in study design, data collection and analysis, decision to publish, or preparation of the manuscript.

==============================
Background

Allelopathic rice releases allelochemicals through its root systems, thereby exerting a negative effect on paddy weeds. This research aimed to evaluate the relationship between fine-root traits and the rice allelopathic potential at the seedling stage.

Methods

Two allelopathic rice cultivars, ‘PI312777’ and ‘Taichung Native1,’ and one non-allelopathic rice cultivar, ‘Lemont,’ were grown to the 3–6 leaf stage in a hydroponic system. Their fine roots were collected for morphological trait (root length, root surface area, root volume, and root tips number) in smaller diameter cutoffs and proliferative trait (root biomass) analysis. Their root-exudates were used for quantitative analysis of phenolic acids contents and an evaluation of allelopathic potential. Correlation analysis was also used to assess whether any linear relationships existed.

Results

Our results showed that allelopathic rice cultivars had significantly higher fine-root length having diameters <0.2 mm, more root tips number, and greater root biomass, coupled with higher allelopathic potential and phenolic acid contents of their root exudates, comparing with non-allelopathic rice cultivar. These fine-root traits were significantly-positively correlated to allelopathic inhibition and total phenolic contents in rice root-exudates. However, there were not significant correlations among the rice allelopathic potential and total phenolic acid contents of rice root-exudates with the root length, root surface area, and root volume of fine root in diameter >0.2 mm.

Discussion

Our results implied that fine-root traits appears to be important in understanding rice allelopathy at the seedling stage. The high allelopathic potential of rice cultivars might be attributed to their higher length of fine roots <0.2 mm in diameter and more number of root tips of fine root, which could accumulate and release more allelochemicals to solutions, thereby resulting in high inhibition on target plants. The mechanisms regulating this process need to be further studied.

Introduction

Allelopathy is the study of interactions between plants which a donor plant release biochemicals (known as allelochemicals) that influence the germination, growth, survival, and reproduction of a receiver plant (Rice, 1984). Root exudation is one of the major pathways of allelochemicals release into environment, which was one of root exudates (Dilday, Nastasi & Smith, 1989; Rice, 1984). The view that allelopathy is related to plant root was first reported in the study of walnut tree where no grass would occur underneath a walnut tree, and death line of the tested plants underneath walnut tree was the same as the growth line of black walnut root (Rietveld, 1983). Juglone was primarily produced by these living walnut roots and was detected in the rhizosphere soil surrounding the living roots (Jose & Gillespie, 1998). Recently, there is no doubt that plant roots and root exudates are the key factors in belowground plant-plant interactions (Bertin, Yang & Weston, 2003; Bais et al., 2004; Baetz & Martinoia, 2014; Tsunoda & Dam, 2017).

The accumulation of root exudates in specialized organs in living roots and their potential role in plant-plant interactions have been documented for several crop and medicinal species. In 2001, Czarnota et al. (2001) described localization and release of sorgoleone by living sorghum root hairs as plant growth inhibitors. Some flavonoids have been shown to mediate allelopathic interactions in the plant rhizosphere, which can accumulate at the root tip of white clover and in root cap cells from where they can be exuded into the soil (Weston & Mathesius, 2013).Through confocal and light microscopic imaging techniques, Zhu et al. (2016) revealed that bioactive naphthoquinones were released by direct exudation in droplets which accumulated at the tips of living root hairs in E. plantagineum. However, there are relatively few studies that have focused on location and release of allelochemicals from plant roots into the rhizosphere of allelopathic rice. Rice allelopathy is first reported in 1990s when Dilday, Nastasi & Smith (1989) found some rice have apparent inhibition effect on neighbouring paddy weeds in field test. They observed that allelopathic rice PI312777 and PI338046 had the greatest root biomass whereas non-allelopathic rice Lemont and M-201 had the least root biomass (Dilday, Mattice & Moldenhauer, 2000). Gealy, Moldenhauer & Duke (2013) quantified the root distribution and potential interactions between allelopathic rice and major weeds using stable 13C isotope discrimination analysis, and the results demonstrated that under weed-free conditions, the roots of the allelopathic cultivars explore the upper soil profile more thoroughly than non-allelopathic cultivars. Thus, we hypothesized there were important relationships between root traits (root tips and root biomass) and rice allelopathy.

Generally, plant root includes meristermatic zone where cell division take place, elongation zone where cells expand along the longitudinal axis, and the differentiation zone where root hairs develop. The root elongation grows by root tip growth that restricted to the growing apex (Datta et al., 2011). Root diameter has a strong influence on root trait variation across plant species, growth forms and biomes (Ma et al., 2018). Fine roots, traditionally defined as all roots ≤ 2 mm in diameter, are primary acquisition organs and mediate biochemical process in terrestrial ecosystems (McCormack et al., 2015). More recently, some studies have assigned increasingly smaller diameter cutoffs (e.g., 1.0 or 0.5 mm) in an effort to explicitly emphasize more functional fine roots (Iversen et al., 2017). Yet it is not clear whether there was a possibility that rice allelopathy was related to root fineness.

To test our hypothesis and clarify the above question, we explored the differential of root traits between allelopathic and non-allelopathic rice at the seedling stages, determined the contents of phenolic acids and allelopathic potential of rice root-exudates, and further analyzed their correlation. Investigating the morphological and proliferative traits of allelopathic rice fine-roots at the seedling stages could help us to understand the formation mechanism of the weed-suppressing zone, and then to extend the weed control area by modifying rice root system.

Materials & Methods

Plant growth, sampling, and root trait calculation

Two internationally recognized allelopathic rice cultivars, ‘PI312777’ (PI) and ‘Taichung Native1’ (TN), and one non-allelopathic rice cultivar, ‘Lemont’ (Le) (Dilday, Mattice & Moldenhauer, 2000), were used in this study. The experiments were conducted in a completely randomized design with three replicates for each cultivar. Surface-sterilized seeds were pregerminated in Petri dishes. Ten uniform seedlings at 1-leaf stage were inserted into the holes of a polystyrene float (27 cm ×17 cm ×1.5 cm), which was floated in a plastic pot (29 cm ×19 cm ×15 cm) containing 5 L of Hoagland’s nutrient solution. Additional distilled water was added to each pot daily to maintain the same volume, and Hoagland’s solution was added to each pot every 7 d. When the rice seedlings were at the 3-, 4-, 5-, and 6-leaf stage, the seedlings were harvested, and their roots were collected for analysis of morphological and proliferation traits.

The fine-root traits were analyzed according to the procedure of Lupini et al. (2016) and Zhang, Fan & Wang (2018). Briefly, three plants of each rice cultivar were randomly selected for root analysis. The clean and intact roots were scanned with an Epson Expression 11000XL scanner (Seiko Epson Co., Nagano-ken, Japan) to yield a grayscale image. The image was processed with WinRHIZO (Regent Instruments Inc., Quebec, Canada) to obtain the following morphological traits: root length, root surface area, root volume in smaller diameter cutoffs (0.06, 0.08, 0.10, 0.2, 0.5, 1.0, and 2.0 mm) of fine root, and root tips number, which were analyzed by WinRHIZO software. And then the roots were oven-dried at 105 °C for 30 min and 80 °C for 48 h to obtain the root biomass.

The rice culture solutions were collected and filtered through Whatman No. 1 filter paper. The filtrate was concentrated to <200 mL by rotary evaporation at 40 °C ± 1 °C, then stored at 4 °C in a refrigerator for 24 h. In this process, the inorganic salts were precipitated and removed by filtration. The filtrate was filtered through a 0.22-µm membrane and diluted to a volume of 200 mL. This solution was used as the concentrated root-exudates for evaluation of the allelopathic potential and quantitative analysis of phenolic acids (Li et al., 2017; Zhang et al., 2018).

Evaluation of allelopathic potential of rice cultivars

The allelopathic potential of the rice cultivars was evaluated using the concentrated root exudates in laboratory bioassays, as described by Zhang et al. (2018). Five milliliters of the concentrated root exudates above at each leaf-stage was added to tissue culture flasks lined with a filter paper at the base, with Hoagland’s solution as a control. Five germinating lettuce (Lactuca sativa L.) seeds were sown on the filter paper of the tissue culture flask that was placed in an incubator (25 ± 2 °C, 12-h light, 6:00–18:00). All of the experiments were repeated three times, and the root length of lettuce was measured after 3 d, and then the plant was oven dried at 105 °C for 30 min and 80 °C for 48 h to obtain the plant dry weight. The % inhibition was used to assess allelopathic potential of three rice cultivars, and was calculated as follows: % inhibition = (1 −T/C) ×100%, based on the root length and plant dry weight of barnyard grass in treatments (T) and in controls (C).

Quantification of phenolic acids in the rice root-exudates

The contents of single phenolic acids in the rice root-exudates were quantified using the external standard method by solid-phase extraction-high-performance liquid chromatography (SPE-HPLC) as described previously (Li et al., 2017). The concentrated root-exudates at each leaf-stage were loaded onto Cleanert PEP solid phase extraction cartridges (Agela Technologies, Tianjin, China). The cartridge was eluted with water and then methanol, and the methanol fraction was concentrated with N2, which was resolved by methanol to 500 µL for quantitative analysis using HPLC.

Quantitative analysis was carried out with an HPLC instrument (HPLC-1260, Agilent, USA) equipped with a C18 reverse-phase column (ZORBAX SB-C18, 150 mm ×4.6 mm, 5 µm) with a UV detector, and phenolic acids were detected at 280 nm. The injection volume of the samples was 5 µL. Elution was performed at a constant flow of 1.6 mL min−1 with methanol (A) and 1% phosphoric acid (B) with the following gradient: initial mobile phase of 27% A at 9 min, then to 30% over 2 min, thereafter increased to 50% for 4 min. The total run time was 15 min. Eight phenolic acids (protocatechuic, p-hydroxybenzoic, vanillic, syringic, p-coumaric, ferulic, salicylic, and cinnamic acid), that widely recognized as rice allelochemicals (Mattice et al., 1998; Seal, Haig & Pratley, 2004; He et al., 2012; Li et al., 2017), were chosen as standards for the calibration curve. The concentrations of single phenolic acids in each rice root-exudates were quantified by interpolating the peak area on the HPLC chromatogram to a standard curve constructed from the peak area of the authentic phenolic acids. The phenolic acid contents were represented as ng per plant.

Statistical analysis

Data were presented as means ± standard error (SE) from three replicates for each experiment or determination. In order to determine any significant differences among treatments, data were analyzed using a two-way analysis of variance (ANOVA) with General Linear Model, followed by Tukey’s honestly significant difference (HSD) tests were performed at P<0.05. To fulfill the assumptions of the ANOVA, transformations were made when required. Pearson’s correlation coefficients (r) between the root morphological traits, root biomass, total phenolic acid contents, and allelopathic potential (% inhibition on plant dry weight of lettuce) were calculated for three rice cultivars at 3–6 leaf stages (n = 12). All of the data analyses were performed with SPSS 20.0 (SPSS Inc., Chicago, IL, USA).

Results

Fine-root traits of allelopathic and non-allelopathic rice cultivars

By the scanning graph, the difference of root growth between three rice cultivars could be obviously observed with the naked eye, including the daughter roots and the adventitious roots. The fine-root traits of allelopathic and non-allelopathic rice cultivars varied in different diameter cutoffs. The results showed that 60–65% of the total fine-root length sampled of allelopathic rice PI and TN can be accounted for by roots that are <0.2 mm in diameter, which was much thinner than that of non-allelopathic rice Le, with 55–60% of total root length having diameters >0.2 mm at 3–6 leaf stages. In order to simplify data for statistical analysis, we chose to analyze the differences of fine-root morphology traits of three rice cultivars in three diameter ranges of 0–0.2 mm, 0.2–1.0 mm and 1.0–2.0 mm. As shown in Fig. 1, the total value of fine-root length in diameter <0.2 mm were all significant higher in allelopathic cultivars PI and TN than that in non-allelopathic cultivar Le at the same stage. Similarly, root tips number and root biomass were significantly higher in PI and TN than that in Le at the same stage (Fig. 2), except not significantly different between TN and Le at the 3- and the 6-stage. However, the total value of fine-root length that are <2 mm in diameter were not significantly different between three rice cultivars at the same stage, and the total value of root surface area and root volume were significantly higher in Le than that in PI and TN (Fig. 1).

Figure 1 Total value of root length (A), root surface area (B), and root volume (C) in the fine-root diameter range of 0–0.2, 0.2–1.0 and 1.0–2.0 mm of three rice cultivars (PI and TN, allelopathic cultivars PI312777 and Taichung Native1; Le, non-allelopathic cultivar Lemont) at 3–6 leaf stages.

Significant differences (P < 0.05) between rice cultivars at the same leaf stage were indicated by different lowercases, and the significant differences for total values of 0–2.0 mm diameter were indicated by different uppercases at the end of the bars, according to Tukey’s honestly test.

Figure 2 Root tips number and root biomass (mg plant−1) of the three rice cultivars (PI and TN, allelopathic cultivars PI312777 and Taichung Native1; Le, non-allelopathic cultivar Lemont) at 3–6 leaf stages.

Significant differences (P < 0.05) between rice cultivars at the same leaf stage were indicated by different lowercases, according to Tukey’s honestly test.

For a given rice cultivars, all morphological traits of fine root increased with leaf stages (Figs. 1 and 2) and these increases were highly significant, as were root biomass traits (Table 1). At seedling stage, the main effect of rice cultivars on fine-root length, root tips number and root biomass was significant (Table 1). Comparing with other fine-root morphological traits, the effect of rice cultivars on the fine-root length less than 0.2 mm in diameter was more significant. But rice cultivars had little effect on root surface area and root volume.

Table 1 The statistical significance of the effects of leaf-stage and cultivars on fine-root traits, total phenolic acid contents, inhibition on root length and plant dry weight of lettuce.

Factor	Dependent variable	df	F	P value	
Leaf stage	root length (0–0.2 mm)	7	114.670	0.001***	
root length (0.2–2.0 mm)	7	10.996	0.001***	
root surface area	7	14.815	0.001***	
root volume	7	12.646	0.011*	
root tips number	7	1088.983	0.002**	
root biomass	7	59.672	0.000***	
total phenolic acid content	7	11.562	0.002**	
inhibition on root length	7	17.104	0.001***	
inhibition on dry weight	7	7.487	0.002**	
Cultivar	root length (0–0.2 mm)	2	18.534	0.014*	
root length (0.2–2.0 mm)	2	1.659	0.048*	
root surface area	2	1.013	0.134	
root volume	2	1.644	0.116	
root tips number	2	135.944	0.040*	
root biomass	2	5.410	0.005**	
total phenolic acid content	2	3.388	0.028*	
inhibition on root length	2	52.911	0.001***	
inhibition on dry weight	2	9.972	0.002**	
Notes.

The multivariate analysis with General Linear Model were used (Fixed Factors: leaf stage, cultivar).

* 0.01 < P < 0.05.

** 0.001 < P < 0.01.

*** P < 0.001.

Allelopathic potential of allelopathic and non-allelopathic rice

In laboratory bioassays, allelopathic potential of rice root-exudates was expressed by % inhibition on root length and plant dry weight of lettuce. As shown in Table 1, leaf stages significantly affected allelopathic potential of root exudates, and the % inhibition was increased with the increase of leaf stages (Fig. 3). At seedling stage, the main effect of rice cultivars on allelopathic potential of rice root-exudates was highly significant (Table 1). Results in laboratory bioassays demonstrated that the root exudates of PI and TN expressed significantly higher inhibition on root length and plant dry weight of lettuce than that of Le at all of 3-6 leaf stages (Fig. 3). The highest % inhibition was at 6-leaf stage, showing 56.30% in PI and 54.25% in TN on root length of lettuce, respectively, and 32.95% in PI and 27.27% in TN on plant dry weight of lettuce, respectively (Fig. 3).

Figure 3 The % inhibition of the root exudates of the three rice cultivars (PI and TN, allelopathic cultivars PI312777 and Taichung Native1; Le, non-allelopathic cultivar Lemont) at 3–6 leaf stages.

Significant differences (P < 0.05) between rice cultivars at the same leaf stage were indicated by different lowercases, according to Tukey’s honestly test.

Phenolic acid contents in the root exudates of allelopathic and non-allelopathic rice

The contents of eight putative phenolic aicds were detected by SPE-HPLC method. The results showed that variation of each phenolic acid in three rice cultivars was inconsistent as the increase of leaf stages (Table 2). The sum of these eight phenolic acids in root exudates was expressed as total phenolic acids contents in root exudates of three rice cultivars. The main effect of leaf stages and rice cultivars on total phenolic acids contents of rice root -exudates was both significant (Table 1). As shown in Fig. 4, total phenolic acids contents were increased with increasing leaf stages, and were significantly higher in PI and TN than that in Le at all of 3–6 leaf stages. Especially, the total contents of the eight phenolic acids reached the maximum at the 6-leaf stage, were 2.36 µg plant−1 in PI and 1.92 µg plant−1 in TN, as well as in 1.16 µg plant−1 Le. It is noteworthy that the contents of cinnamic acid in PI and TN were 8-times and 6-times higher than that in Le, respectively (Table 2).

Table 2 The contents of eight phenolic acids in rice root-exudates at 3–6-leaf stage (ng plant−1).

Phenolic acid	cultivars	3-leaf	4-leaf	5-leaf	6-leaf	
Protocatechuic acid	PI	10.58 ± 0.86	30.49 ± 5.45	14.62 ± 1.13	26.82 ± 2.40	
TN	9.36 ± 0.36	25.85 ± 1.59	14.33 ± 1.25	24.09 ± 2.52	
Le	11.63 ± 0.52	22.37 ± 1.32	26.74 ± 1.24	ND	
p- Hydroxybenzoic acid	PI	41.43 ± 1.12	674.65 ± 9.89	49.46 ± 3.24	66.56 ± 2.84	
TN	36.4 ± 2.86	566.4 ± 5.26	38.4 ± 2.31	68.55 ± 3.01	
Le	52.43 ± 1.14	71.23 ± 3.09	17.21 ± 1.22	28.77 ± 1.93	
Vanillic acid	PI	15.74 ± 0.99	64.14 ± 6.26	43.84 ± 6.67	78.37 ± 4.62	
	TN	14.54 ± 0.82	59.39 ± 4.37	39.79 ± 3.14	74.14 ± 4.84	
	Le	16.14 ± 1.53	12.47 ± 3.25	13.18 ± 2.29	10.99 ± 1.68	
Syringic acid	PI	59.65 ± 3.28	43.06 ± 2.46	111.09 ± 5.34	282.73 ± 7.96	
	TN	56.66 ± 3.76	36.87 ± 3.17	109.38 ± 6.29	315.89 ± 7.15	
	Le	52.66 ± 1.46	174.05 ± 6.24	102.95 ± 6.21	95.42 ± 2.36	
p-Coumaric acid	PI	ND	ND	ND	ND	
	TN	9.45 ± 0.21	16.37 ± 0.75	13.41 ± 0.25	48.54 ± 1.68	
	Le	ND	ND	ND	ND	
Ferulic acid	PI	9.71 ± 0.11	19.48 ± 1.21	14.67 ± 1.08	53.74 ± 3.12	
	TN	ND	ND	ND	ND	
	Le	ND	ND	ND	23.87 ± 1.18	
Salicylic acid	PI	252.54 ± 7.12	499.58 ± 8.02	1176.52 ± 11.89	1818.59 ± 15.14	
	TN	207.23 ± 7.36	483.55 ± 6.93	1138.91 ± 12.42	1362.09 ± 10.22	
	Le	140.99 ± 4.37	303.08 ± 8.14	888.29 ± 19.51	996.66 ± 9.78	
Cinnamic acid	PI	1.90 ± 0.12	15.10 ± 0.31	33.59 ± 1.78	37.91 ± 2.13	
	TN	1.54 ± 0.14	14.44 ± 0.93	31.56 ± 2.64	30.06 ± 2.78	
	Le	1.37 ± 0.38	6.79 ± 0.59	10.12 ± 2.09	4.43 ± 0.26	
Notes.

Means ± standard error (SE) from three replications for each determination is shown. ‘ND’means compounds not detected in the analysis. PI and TN, allelopathic cultivars PI312777 and Taichung Native1. Le, non-allelopathic cultivar Lemont.

Figure 4 Total contents of eight phenolic acids in rice root-exudates at 3–6-leaf stage.

Significant differences (P < 0.05) between rice species at the same leaf stage were indicated by different lowercases according to Tukey’s honestly test. PI and TN, allelopathic cultivars PI312777 and Taichung Native1. Le, non-allelopathic cultivar Lemont.

Correlation between root traits and allelopathic potential

As shown in Table 3, allelopathic potential was significantly correlated with the total phenolic acid contents of rice root-exudates (r = 0.914, P < 0.001), fine-root length at 0–0.2 mm (r = 0.906, P < 0.001), root tips number (r = 0.921, P = 0.001), and root biomass (r = 0.843, P = 0.001), but was not significantly correlated with fine-root length at 0.2–2.0 mm, root surface area, and root volume. The total phenolic acids contents of rice root-exudates was significantly correlated with fine-root length at 0–0.2 mm (r = 0.958, P < 0.001), root tips number (r = 0.937, P < 0.001), and root biomass (r = 0.895, P < 0.001), but was not significantly correlated with fine-root length at 0.2–2.0 mm, root surface area, and root volume. Besides, the fine-root length at 0–0.2 mm was significantly correlated with root tips number (r = 0.972, P < 0.001) and root biomass (r = 0.946, P < 0.001), and root tips number was significantly correlated with root biomass (r = 0.928, P < 0.001). In summary, there were significant correlations among rice allelopathic potential and total phenolic acid contents of rice root-exudates with fine- root length at 0–0.2 mm, root tips number, and root biomass.

Table 3 Correlation coefficients (r) among total phenolic acids contents of rice root-exudates, rice allelopathic potential andfine- root traits of three rice cultivars.

		Plant	Root morphological trait	
		RAP	PAC	RL(1)	RL(2)	RTN	RSA	RV	
Plant	RAP	/							
PAC	0.914***	/						
Root morphological trait	RL(1)	0.906***	0.958***	/					
RL(2)	0.428	0.477	0.568	/				
RTN	0.921***	0.937***	0.972***	0.437	/			
RSA	0.451	0.406	0.599	0.991***	0.447	/		
RV	0.249	0.407	0.496	0.953***	0.429	0.962***	/	
Root proliferative trait	RB	0.843**	0.895**	0.946***	0.825**	0.928***	0.848**	0.698*	
Notes.

RAP- rice allelopathic potential(% inhibition on plant dry weight of lettuce), PAC- total phenolic acids contents of rice root-exudates, RL(1)- root length at 0-0.2mm, RTN- root tips number, RL(2)- root length at 0.2–2.0 mm, RSA- root surface area, RV- root volume, RB- root biomass.

* 0.01 <  P < 0.05.

** 0.001 <  P < 0.01.

*** P < 0.001.

Discussion

Plant roots play important roles in belowground interactions with neighboring plants for space, water, and nutrients, and with soil microbes (Bais et al., 2004; Callaway & Mahall, 2007; Mallik, Biswas & Collier, 2016; Laliberté, 2017). On one hand, root traits, including morphology, distribution, architecture, and biotic community influence plant growth, function and biochemical processes (Bardgett, Mommer & De Vries, 2014; Laliberté, 2017). On the other hand, plant root can synthesize, accumulate, and secrete a diverse array of compounds, referred to as root exudates that impact the soil microbial community, change the physio-chemical properties of soil, and inhibit the growth of competing plant species (Bertin, Yang & Weston, 2003; Bais et al., 2004; Baetz & Martinoia, 2014; Tsunoda & Dam, 2017; Rovira, 1969; Walker et al., 2003).

Our hypothesis proposed there were important relationships between root traits (root tips and root biomass) and rice allelopathy. This proved to be true for allelopathic rice regardless of leaf stages. Our results clearly showed that two allelopathic rice cultivars PI and TN had significantly higher more root tips number, and greater root biomass, coupled with higher allelopathic inhibition and phenolic contents of their root-exudates, comparing with non-allelopathic rice Le at the allelopathic stage (i.e., at 3–6 leaf stages) in a hydroponic system (Figs. 1–4). All results were in agreement with most previously reported results. Dilday, Mattice & Moldenhauer (2000) first previously reported that allelopathic rice cultivars had the greatest root biomass than non-allelopathic rice cultivars in field test. Until recently, Gealy et al. also observed that the root distribution of allelopathic rice cultivars is shifted towards the upper soil with relatively greater root biomass in comparison to non-allelopathic rice cultivars (Gealy & Moldenhauer, 2012; Gealy, Moldenhauer & Duke, 2013). In addition, these root traits were significantly-positively correlated to allelopathic inhibition and total phenolic contents in rice root-exudates (Table 3), which has not been reported before. It seems that more root tips and root biomass might help the exudation accumulation in root tips, and is similar to that observed in living E. plantagineum roots (Zhu et al., 2016). However, we need to further explore the mechanism of localization and release of allelochemicals from rice roots into the rhizosphere.

The root system of a rice plant consists of seminal root originated from the seed and adventitious roots formed from the stem. Formation, emergence, elongation and branching of adventitious roots proceed along the main stem, to keep pace with successive leaf stage (Morita & Abe, 1993). As observed in fine roots of nine American trees by Pregitzer et al. (2002), roots <0.2 mm in diameter would account for the majority of the fine roots in two allelopathic rice cultivars regardless of leaf stages. Moreover, fine-root length <0.2 mm in diameter were all significantly higher in allelopathic cultivars PI and TN than that in non-allelopathic cultivar Le at the same stage (Fig. 1). Furthermore, there were significant correlations among the rice allelopathic potential and total phenolic acid contents of rice root-exudates with fine-root length at 0–0.2 mm in diameter. However, there were not significant correlation relationships among the rice allelopathic potential and total phenolic acid contents of rice root exudates with the length of fine root >0.2 mm in diameter, root surface area, and root volume (Table 3). All this indicates that the most reasonable fine root size class for allelopathic rice we studied would consist of all roots smaller size in diameter than non-allelopathic rice. Moreover, the longer the length of fine roots less than 0.2 mm in diameter, the more likely allelochemicals are to be released and diffused into the soil, which needs to be further explored.

Rice allelopathy is involved in the chemical-mediated interaction between rice and neighbouring weeds (He et al., 2012; Kato-Noguchi & Ino, 2013; Zhang et al., 2018). Allelopathic rice expressed suppressive- activity on weeds when the plant from the 3- to 7-leaf stages and reached the maximum at 5-6 leaf stages (Li et al., 2015). Large quantities of root exudates, including phenolic acids, were typically released from living root hairs (Baetz & Martinoia, 2014; Bertin, Yang & Weston, 2003; Lynch, 1995). In this paper, we once again proved that there were significant differences in contents of phenolic acid in root exudates between allelopathic and non-allelopathic rice at seedling stage, which was consistent with a previous report (Li et al., 2017). Above all, the differences were significantly correlated with fine-root traits and allelopathic potential of rice roots. These results implied that there was a close linkage between the structural attributes of fine roots and how they control the weed. The high allelopathic potential of rice cultivars might be attributed to their higher length of fine roots <0.2 mm in diameter and more number of root tips of fine root, which could accumulate and release more allelochemicals to solutions, thereby resulting in high inhibition on target plants. However, the mechanisms regulating this process need to be further studied.

Research has confirmed that modifying crop root systems could increase crop yield, absorb and utilize more soil nutrients, regulate water under drought or waterlogging stresses, as well as recognize the neighbour plants (Asaduzzamana et al., 2016; Asch et al., 2005; DuPont et al., 2014; Jeong et al., 2013; Jitsuyama, 2015; Uga et al., 2013; Zhang et al., 2009). Using a genetic approach, Uga et al. (2013) successfully altered the growth angle of rice roots in a more downward direction and increased deep rooting, which enabled rice to avoid drought and maintain high yield under drought conditions. Similarly, our results open a novel way for modifying the distribution and architecture of the rice root system to enhance rice allelopathy, which needs to be further studied under field conditions.

Conclusions

In this study we found that allelopathic rice cultivars had significantly greater length of fine roots <0.2 mm in diameter, more root tips number and greater root biomass of fine root, comparing with non-allelopathic rice cultivar at 3–6 leaf stages in a hydroponic system. These fine-traits were highly correlated with allelopathic potential and the phenolic acid contents in rice root-exudates. These findings provide new evidence that rice allelopathy can be enhanced by altering rice root system, and would be useful for weed control in rice fields.

Supplemental Information

Supplemental Information 1 Raw data exported from fine-root traits, allelopathic potential and phenolic acid contents in the root exudates of three rice cultivars

(PI and TN, allelopathic cultivars PI312777 and Taichung Native1. Le, non-allelopathic cultivar Lemont) at 3–6 leaf stages applied for data analyses and preparation for Figs. 1–4.

Click here for additional data file.

We acknowledge Professor Xinyu Zheng and Dr. Jiebo Chen for their helpful suggestions.

Additional Information and Declarations

Competing Interests

Author Contributions

Data Availability

The authors declare there are no competing interests.

Jiayu Li conceived and designed the experiments, analyzed the data, prepared figures and/or tables, approved the final draft.

Shunxian Lin and Qingxu Zhang performed the experiments.

Qi Zhang analyzed the data.

Wenwen Hu contributed reagents/materials/analysis tools.

Haibin He conceived and designed the experiments, authored or reviewed drafts of the paper, approved the final draft.

The following information was supplied regarding data availability:

The raw data are available in the Supplemental Files.

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
