# Peer review of "Fine-root traits of allelopathic rice at the seedling stage and their relationship with allelopathic potential"

_PeerJ, doi:10.7717/peerj.7006_

## Round 0.1 · original submission · Major Revisions

The manuscript has been reviewed by two experts in the filed. Both the reviewers raise substantial concerns regarding the manuscript as indicated in their detailed report. Reviewer 2 has been more critical than Reviewer 1. I am willing to consider a revised manuscript that addresses all of the concerns of both the reviewers. I agree with reviewer # 2 that the English language needs to be significantly improved in order for the manuscript to be accepted.

Reviewer 1 ·

Basic reporting

The current article has a sound experimental design and clear results, while a more sufficient research background, a thorough discussion, and a less speculate conclusion are needed.

Experimental design

The experiments design was simple to follow. However, working hypotheses were not dressed clearly.

The authors should give sufficient background why the definition of fine roots in
the current article is 0.2mm, which is not common in this field.
Experiment methods should have citations to show its validity. If the authors inserted any modification, you should also describe where and why the original protocol had been modified.

The statistical analysis focused only on species-variation but not development stages, e.g. data in table one are suitable for two-way ANOVA.

Validity of the findings

1. The correlation between root traits and allelopathy potential is novel and interesting.
2. The author should pay attention that neither the previous studies nor the current study probably indicated any causality between these factors.

3. Most of the results were clear and rigorous except that the comparison of total phenolic acids among allelopathic rice species and non-allelopathic one which I would like to suggest a comparison not on whole plant base but root mass (or length) base.

4. The discussion on the correlation between root traits and allelopathy potential is not sufficient. Possible mechanisms and significance of this correlation should be included in the discussion.

5. The function of each phenolic acids was not touched in the discussion while the authors reported the species variation of phenolic acids in detail.

6. The authors apparently ignored the variations of root traits among allelopathic rice and non-allelopathic one, which also have interesting implications. For example, the specific root-biomass of Le was highest while its surface area was low, in another word, the specific root area of Le was low. Thus, on the root-area-base, non-allelopathic rice had even less impact on its surrounding.

Additional comments

The finding of this article is novel and will contribute new information to the root traits area.
However, I would like to suggest the authors change the display of their current results to the take-home-message easy to follow.
Make a less speculate conclusion about the correlations between fine root traits and allelopathy potential.
The introduction and discussion of the current article were not well-structured to provide background for the common audience to follow the scientific significance.
Please see annotated PDF for more detailed suggestions.

Annotated reviews are not available for download in order to protect the identity of reviewers who chose to remain anonymous.

Reviewer 2 ·

Basic reporting

The manuscript is not written very well. English language used is not professional. For example, Line 54, previous studies have shown that..... Line 65, Rice yield can be increased by controlling the direction of root extension. Authors should pay more attention to English language. One of the effective ways is reading more papers.
The authors tried to tell a story about the relationship between root traits (such as morphological and root growth) and the potential weed-suppressive ability. In general, authors have cited few important references about the potential weed-suppressive ability. However, references about root traits are not sufficient.
The structure is not good.The introduction is very hard to follow so that I can not get a clearer picture in my mind of the story that authors want to tell. There are many issue in figures and tables. For instance, I can not separate thinner (or fine) roots with the range of 0~0.2mm from thicker (or coarse) roots with the range of 1.0~2.0mm.
Authors did not show the ANOVA results about the relationship between root traits and the total contents of eight phenolic acids or I missed. More importantly, authors expressed so-called weed-suppressive ability as the total contents of phenolic acids rather than the inhibition effect. authors should analyse the relationship between root traits and the inhibition effect.

Experimental design

This original work is within the scope of the journal. However, authors do not explain the significance of their work. People know that the allelopathic compounds are very hard to measure. If there are good relationships between root traits and allelopathic compounds, people can predict the content with root traits. On the other hand, people can select varieties for the green agriculture. Authors should mention this.
Authors did not describe the experimental design very well, especially the inhibition effect using root exudates. Authors should describe methods with many important references and sufficient detail. For instance, authors should show clearly how many plant roots were scanned, one or all per pot.

Validity of the findings

According to the introduction authors wrote, few studies investigated the relationship between root traits (root distribution and biomass) and the potential weed-suppressive ability. This work could improve our understanding of the association of the potential weed-suppressive ability with root morphological traits. However, there are serval issues about statistics. Authors should show the ANOVA results (p values and method such as Pearson correlation) as a table or figure in the text to detect the linkage between root traits and phenolic acids. Authors also should detect the linkage between root traits and the inhibition effect which is an important data to support the conclusion.

Additional comments

In the introduction, authors should give us some information about fine root traits of allelopathic rice because they analysed their effect on the inhibitory effect on weeds in the following text.
Authors should correct the technical issues in the text and references carefully, such as Line 120, Line 312, 314, etc.

---

## Round 0.2 · Major Revisions

One of the reviewers is still not happy with the revision. I invite you to re-submit the manuscript after addressing the reviewer's concerns. I agree with the reviewer that the language should be improved.

Reviewer 1 ·

Basic reporting

The manuscript needs thorough grammar editing. Even the abstract had several mistakes.
The authors cited some of the most recent researches but failed to construct own hypotheses based on them.
The description of the results was very hard to follow, especially those in Table 1 and 2.
The discussion was extremely simple and did not match the introduction. Many topics and significance raised in the introduction were not mentioned again in the discussion.

Experimental design

The experimental methodology was sound.
Yet, there's no clear testable hypothesis for the experimental design.

Validity of the findings

The main findings are interesting and the authors made no speculate conclusions.

However, some of the results lacked proper statistical analysis.
The description of the output of a two-way ANOVA is very hard to follow.

Additional comments

Introduction
The current introduction did not explain the research purpose well.
The introduction ought to address why there are possible correlations between root traits and allelochemical potential.
The aims of this research laced a clear hypothesis. In this research, why root traits are important to the release pattern of root exudates to the surrounding soils should be the take-home message.

Material and methods
Why is the average value of root diameter used as a category standard of the fine roots?
Based on Figure S1, the distribution of the raw diameter data of all scanned fine roots should not be a normal distribution but a skewed one, which means the proportions of adventitious roots were different among varieties so that average value may not be an informative parameter for a group of roots not along the definition of thinner roots in this research.

Results
The result part should be improved.
The description of Table 1 was too hard to follow. I think the author probably interpreted the two-way ANOVA in a wrong way.
There's no statistic result supporting these statements in Fig. 4 or any other part of this manuscript!

Discussion
The authors should describe why the current research is needed and how does it relate with the previous researches listed here.

Please see the details in the annotated PDF.

Annotated reviews are not available for download in order to protect the identity of reviewers who chose to remain anonymous.

---

## Round 0.3 · accepted · Accept

You have successfully addressed all the concerns raised by the reviewer and I am happy to accept the manuscript for publication in PeerJ.

# This was a relatively simple experiment and it would have been nicer if a larger germplasm collection were used to observe the differences. On the surface, the statistical impact is marginal with the small sample size; however, the study appeared to sufficiently match the criteria needed for publication and may serve as a pilot study for further evaluation.

I also include are some further editing suggestions:

EDITS LINE NO: / BEFORE / AFTER / [COMMENTS]
LINE 25: / phenolic acids contents / phenolic acid content / [.]
LINE 30: / acid contents / acid content / [.]
LINE 30: / non-allelopathic rice / the non-allelopathic rice / [.]
LINE 31: / phenolic contents / phenolic content / [.]
LINE 32: / phenolic acid contents / phenolic acid content / [.]
LINE 37: / target plants / target weed plants / [.]
LINE 67: / includes meristermatic / includes a meristematic / [.]
LINE 69: / that restricted / that is restricted / [.]
LINE 105: / bioassay, / bioassays, / [.]
LINE 105: / exudates above / exudates highlighted above / [.]
LINE 125: / widely / are widely / [.]
LINE 132: / tests were / tests / [.]
LINE 144: / make the data easy / simplify data / [.]
LINE 144: / we choose to / we chose to / [.]
LINE 147: / in non-allelopathic / in the non-allelopathic / [.]
LINE 148: / were significant / were significantly / [.]
LINE 148: / significant difference / significantly different / [.]
LINE 151: / significant higher / significantly higher / [.]
LINE 153: / given rice cultivars / given rice cultivar / [.]
LINE 160: / laboratory bioassay / laboratory bioassays / [.]
LINE 164: / bioassay / bioassays / [.]
LINE 165: / of % inhibition / % inhibition / [.]
LINE 172: / acids contents / acid content / [.]
LINE 173: / acids contents / acid content / [.]
LINE 174: / acids contents were / acid content was / [.]
LINE 175: / were significantly / was significantly / [.]
LINE 175: / Specially, / ? / [Especially or Specifically? Reword sentence.]
LINE 184: / phenolic acids contents / phenolic acid content / [.]
LINE 187: / root tips number / root tip number / [.]
LINE 188: / root tips number / root tip number / [.]
LINE 200: / was there / proposed there / [.]
LINE 227: / allelopathy involved / allelopathy is involved / [.]
LINE 233: / with previous / with a previous / [.]
LINE 235: / linkages / linkage / [.]
LINE 239: / A lot of researches / Research has / [.]
LINE 239: / utilize of / utilize more / [.]
LINE 242: / By genetic approach / Using a genetic approach / [.]
LINE 242: / successful altered / successfully altered / [.]
LINE 243: / maintained high / maintain high / [.]
LINE 244: / of rice root system / of the rice root system / [.]
#